# Inhibiting host-protein deposition on urinary catheters reduces associated urinary tract infections

**Marissa Jeme Andersen[1], ChunKi Fong[2,3], Alyssa Ann La Bella[1], Jonathan Jesus Molina[1], Alex Molesan[1], Matthew M Champion[4], Caitlin Howell[2,3]\*, Ana L Flores-Mireles[1]\***

[1]Department of Biological Sciences, College of Science, University of Notre Dame, Notre Dame, United States; [2]Department of Chemical and Biomedical Engineering, College of Engineering, University of Maine, Orono, United States; [3]Graduate School of Biomedical Science and Engineering, University of Maine, Orono, United States; [4]Department of Chemistry and Biochemistry, College of Science, University of Notre Dame, Notre Dame, United States

**Abstract** Microbial adhesion to medical devices is common for hospital-acquired infections, particularly for urinary catheters. If not properly treated these infections cause complications and exacerbate antimicrobial resistance. Catheter use elicits bladder inflammation, releasing host serum proteins, including fibrinogen (Fg), into the bladder, which deposit on the urinary catheter. *Enterococcus faecalis* uses Fg as a scaffold to bind and persist in the bladder despite antibiotic treatments. Inhibition of Fg–pathogen interaction significantly reduces infection. Here, we show deposited Fg is advantageous for uropathogens *E. faecalis*, *Escherichia coli*, *Pseudomonas aeruginosa*, *K. pneumoniae*, *A. baumannii*, and *C. albicans*, suggesting that targeting catheter protein deposition may reduce colonization creating an effective intervention for catheter-associated urinary tract infections (CAUTIs). In a mouse model of CAUTI, host-protein deposition was reduced, using liquid-infused silicone catheters, resulting in decreased colonization on catheters, in bladders, and dissemination in vivo. Furthermore, proteomics revealed a significant decrease in deposition of host-secreted proteins on liquid-infused catheter surfaces. Our findings suggest targeting microbial-binding scaffolds may be an effective antibiotic-sparing intervention for use against CAUTIs and other medical device infections.

\*For correspondence:
caitlin.howell@maine.edu (CH);
afloresm@nd.edu (ALF-M)

**Competing interest:** The authors declare that no competing interests exist.

## Editor's evaluation

We assessed your work and were impressed by the number of different experiments and the coherence between them, clearly demonstrating the potential of your work for future prevention of catheter-associated urinary tract infections.

## Introduction

Urinary catheters drain patient's bladders during surgical sedation and recovery in addition to being used in treatment for a variety of chronic conditions, making it an exceedingly common procedure in healthcare facilities. Despite the benefits urinary catheters provide for patients, catheterization causes adverse effects including infections and bladder stones (*Andersen and Flores-Mireles, 2019*; *Feneley et al., 2015*). The most common complication is catheter-associated urinary tract infection (CAUTI), which accounts for 40% of all hospital-acquired infections (*Andersen and Flores-Mireles,*

2019; *Feneley et al., 2015*). Unfortunately, CAUTIs often lead to bloodstream infections and systemic dissemination with a 30% mortality rate, causing significant financial burdens for hospitals and patients (*Andersen and Flores-Mireles, 2019*; *Feneley et al., 2015*).

New and current management guidelines for CAUTIs have resulted in moderate reductions in incidences (*Assadi, 2018*; *Meddings et al., 2014*). The standard treatment for patients with symptomatic CAUTI is catheter removal and replacement and an antibiotic regiment (*Assadi, 2018*; *Flores-Mireles et al., 2019*). However, this approach is not effective because biofilms on the catheter surface and bladder wall protect microbes against antibiotics and the immune system and creates the potential development of antimicrobial resistance (*Flores-Mireles et al., 2019*; *Trautner and Darouiche, 2004*). Many ongoing efforts to prevent biofilms on catheters are focused on developing surface modifications (*Andersen and Flores-Mireles, 2019*; *Faustino et al., 2020*; *Singha et al., 2017*). Catheters impregnated with antimicrobials, such as metal ions and antibiotics, have become popular and are now commercialized due to promising in vitro work but, in clinical trials these catheters have shown, at best, mixed results (*Andersen and Flores-Mireles, 2019*; *Singha et al., 2017*). Importantly, there is concern that this approach may not be a long-term solution given that the presence of antimicrobial compounds may drive development of resistance (*Singha et al., 2017*; *Westfall et al., 2019*). Especially when considering the host factors that coat urinary catheters, which could potentially inhibit or decrease pathogen interaction with antimicrobials.

Recent findings in mouse and human urinary catheterization have unveiled the importance of the host clotting factor 1, fibrinogen (Fg), for surface adhesion and subsequent establishment of biofilms and persistence of CAUTIs in *Enterococcus faecalis* and *Staphylococcus aureus* infections (*Flores-Mireles et al., 2019*; *Flores-Mireles et al., 2014*; *Flores-Mireles et al., 2016a*; *Flores-Mireles et al., 2016b*; *Gaston et al., 2020*; *Klein and Hultgren, 2020*). Fg is continuously released into the bladder lumen in response to mechanical damage to the urothelial lining caused by catheterization (*Flores-Mireles et al., 2019*; *Flores-Mireles et al., 2014*; *Flores-Mireles et al., 2016a*; *Klein and Hultgren, 2020*). Once in the lumen, Fg is deposited on the catheter, providing a scaffold for these incoming uropathogens to bind and establish infection in human and mouse CAUTI. When blocking the interaction between Fg and *E. faecalis* using antibodies, the pathogen is not able to effectively colonize the bladder (*Di Venanzio et al., 2019*; *Flores-Mireles et al., 2014*; *Flores-Mireles et al., 2016a*; *Gaston et al., 2020*; *Walker et al., 2017*).

Thus, we hypothesized that reducing availability of binding scaffolds, in this case Fg, would decrease microbial colonization in a catheterized bladder. To test our hypothesis, we used a mouse model of CAUTI and a diverse panel of uropathogens, including *E. faecalis*, *C. albicans*, uropathogenic *Escherichia coli*, *Pseudomonas aeruginosa*, *A. baumannii*, and *Klebsiella pneumonia*, which we found all bind more extensively to catheters with Fg present. To resolve the deposition of Fg, we focus on antifouling modifications, specifically, liquid-infused silicone (LIS). LIS is more simple to make, more stable, and more cost effective than other antifouling polymer modifications (*Andersen and Flores-Mireles, 2019*; *Campoccia et al., 2013*; *Homeyer et al., 2019*; *Howell et al., 2018*; *Chamy, 2013*; *Singha et al., 2017*; *Sotiri et al., 2018*; *Villegas et al., 2019*). Additionally, LIS has shown to reduce clotting in central lines and infection in skin implants (*Chen et al., 2017*; *Leslie et al., 2014*). We show that our LIS-catheters reduced Fg deposition and microbial binding not only in vitro but also in vivo. Furthermore, LIS-catheters significantly decrease host-protein deposition when compared to unmodified (UM)-catheters as well as reducing catheter-induced inflammation. These findings suggest that targeting host-protein deposition on catheter surfaces and the use of LIS-catheters are plausible strategies for reducing instances of CAUTI.

## Results
### Uropathogens interact with Fg during CAUTI

Due to the understood interaction between Fg and some uropathogens and Fg accumulation on catheters over time in human and mice, we assessed potential interaction of *E. faecalis* OG1RF (positive control) uropathogenic *E. coli* UTI89, *P. aeruginosa* PAO1, *K. pneumoniae* TOP52, *A. baumannii* UPAB1, and *C. albicans* SC5314 with Fg in vivo, using a CAUTI mouse model, which recapitulates human CAUTI pathophysiology (*Flores-Mireles et al., 2019*; *Flores-Mireles et al., 2014*; *Flores-Mireles et al., 2016a*; *Flores-Mireles et al., 2016b*). Mice catheterized and infected with

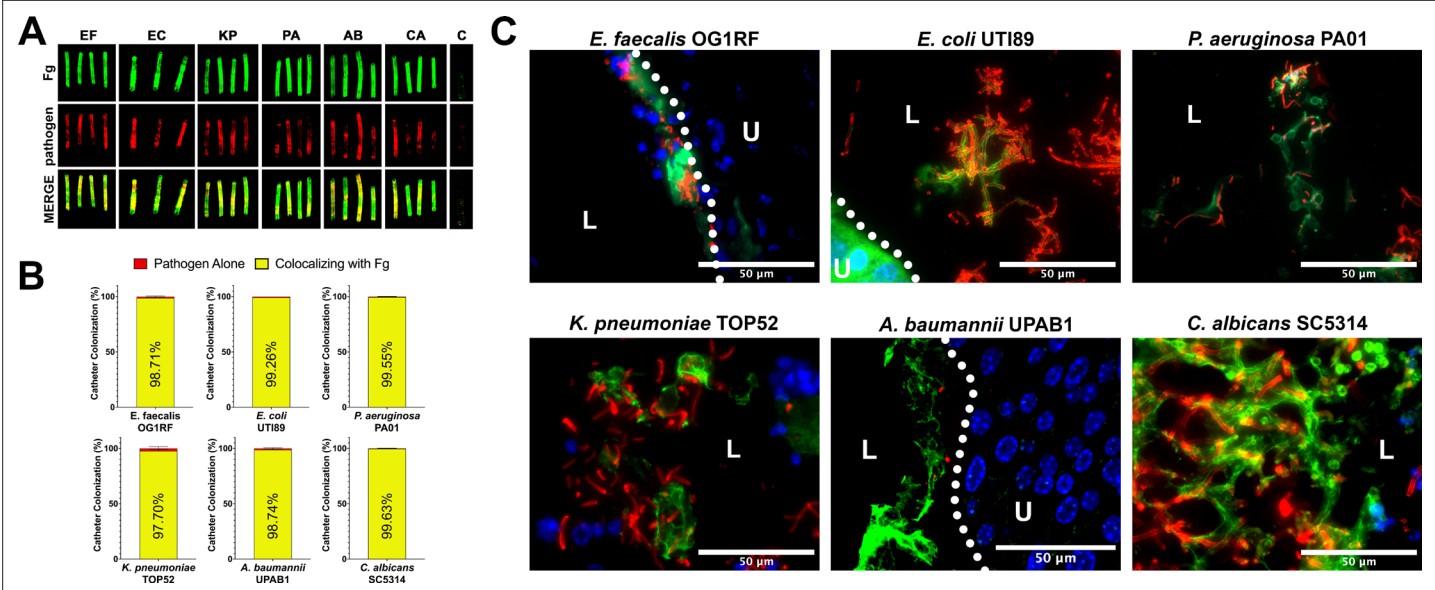

**Figure 1.** Uropathogens interact with fibrinogen (Fg) in vivo. (**A**) Urinary catheters stained with immunofluorescence (IF) for Fg deposition (Fg; green) and microbe binding (respective pathogen; red). Unimplanted catheters were used as controls for autofluorescence, *n* = 3–4. (**B**) Quantification of uropathogen–Fg colocalization on catheters from panel A. (**C**) Representative images from a single bladder illustrating the interaction of uropathogens (red), Fg (green), and nuclei (blue) on the bladder urothelium (U) and in the lumen (L). Scale bar, 50 nm. Montages can be found in *Figure 1—figure supplement 1*. For all graphs error bars show the standard error of the mean (SEM). Between 3 and 5 replicates of *n* = 4–12 each were performed for each pathogen and condition.

The online version of this article includes the following figure supplement(s) for figure 1:

**Figure supplement 1.** Montages of *Figure 1* merged images.

the respective uropathogen were sacrificed at 24 hours post infection (hpi). Catheters and bladders were harvested, stained, and imaged. Visual and quantitative analysis of the catheters showed all uropathogens colocalizing strongly with Fg deposits exhibiting preference for Fg (*Figure 1A, B*) and robust Fg deposition on catheters, validating previous studies on human catheters (*Flores-Mireles et al., 2019*; *Flores-Mireles et al., 2014*; *Flores-Mireles et al., 2016a*; *Flores-Mireles et al., 2016b*). Importantly, immunofluorescence (IF) analysis of bladder sections showed that all uropathogens interact with Fg on the bladder urothelium or in the lumen during CAUTI (*Figure 1C* and montages in *Figure 1—figure supplement 1*). Although we show interaction between the pathogens and Fg, further studies are needed to characterize each pathogen–Fg interaction mechanism, as previously done with *E. faecalis* and *S. aureus* (*Flores-Mireles et al., 2014*; *Flores-Mireles et al., 2016b*; *Walker et al., 2017*).

## Fg on urinary catheter material enhances microbial binding

Based on our in vivo findings, we assessed whether Fg could promote initial binding of the uropathogens to silicone catheters as previously seen for *E. faecalis* (*Flores-Mireles et al., 2014*). In addition to Fg, bovine serum albumin (BSA) was tested since serum albumin is one of the most abundant proteins on human and mouse urinary catheters (Molina et al., in preparation; *Supplementary file 2*). We compared uropathogen binding to Fg-, BSA-, and uncoated silicone, finding that Fg significantly enhanced the binding to the catheter for all uropathogens when compared with uncoated and BSA-coated silicone catheters (*Figure 2*). Interestingly, *P. aeruginosa* and *A. baumannii* binding to BSA-coated silicone was ~14% and ~10% higher than uncoated controls, respectively (*Figure 2C, E*), alluding to a role for other host-secreted proteins during infection. However, these values were still significantly lower than the increase in binding observed on Fg-coated silicone (*Figure 2C, E*). Taken together, these data suggest that uropathogen interaction with host proteins deposited on silicone surfaces, particularly Fg, increases the ability of uropathogens to colonize urinary catheters.

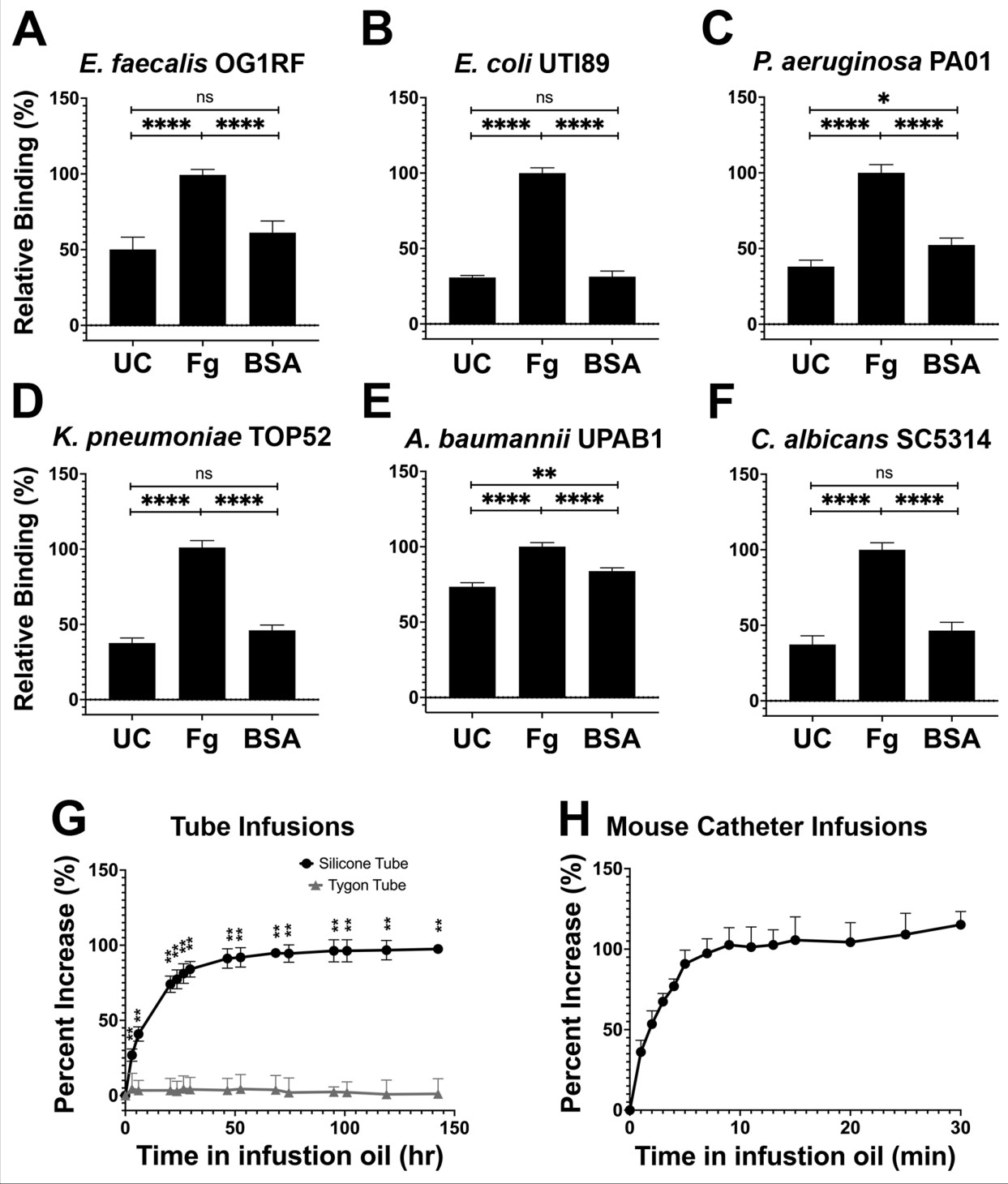

**Figure 2.** Silicone infusion and fibrinogen (Fg) enhancement of microbial surface binding. (**A–F**) Uropathogens were tested for their ability to bind to protein coated and uncoated (UC) silicone catheters. For all graphs, error bars show the standard error of the mean (SEM). Between 3 and 5 replicates of *n* = 4–12 each were performed for each pathogen and condition. (**G**) Kinetics of silicone oil infusion on silicone and Tygon tubes, as well as (**H**) mouse silicone catheters. Differences between groups were tested for significance using the Mann-Whitney U test. *, P < 0.05; **, P < 0.005; and ****, P < 0.0001.

The online version of this article includes the following figure supplement(s) for figure 2:

**Figure supplement 1.** Characterization of LI tubing and mouse catheters.

## Characterization of liquid-infused catheters to prevent host-protein deposition

Based on the exploitative interaction of uropathogens with deposited Fg, we hypothesized that development of a material to prevent protein deposition would also reduce microbial colonization. Recent work with liquid-infused surfaces have demonstrated resistance to protein and bacterial fouling (*Goudie et al., 2017*; *Howell et al., 2018*; *Leslie et al., 2014*; *Sotiri et al., 2018*). This prompted us to develop a LIS material by modifying medical-grade silicone using inert trimethyl-terminated polydimethylsiloxane fluid (silicone oil) (*Goudie et al., 2017*). Infusion was completed by submerging silicone into medical-grade silicone oil, the oil was then naturally taken up by the silicone tube creating a fully infused silicone tube with a slippery surface. Analysis of the oil's infusion rate showed a significant increase in silicone weight during the first 3 days of infusion then a gradual decrease in infusion until a plateau was reached after ~50 hr (raw weight in *Figure 2G* and *Figure 2—figure supplement 1A*). Plastic Tygon tubes (nonsilicone) were used as negative controls (*Figure 2—figure supplement 1B*). Full infusion of mouse silicone catheters was achieved by 10 min of infusion (*Figure 2H* and *Figure 2—figure supplement 1C*). Investigation of silicone tube dimensions showed an increase in length, outer diameter, and inner diameter of ~41.3%, ~ 103.1%, and ~27.6%, respectively (*Figure 2—figure supplement 1D*) and mouse catheters showed an increase of ~30.7%, ~28.7%, and ~39.8%, respectively (*Figure 2—figure supplement 1E*). Based on these results, to ensure full infusion for further assays, silicone tubing was submerged in silicone oil for a minimum of 5 days and for mouse catheters for a minimum of 30 min.

## LIS modification reduces Fg deposition and microbial-binding in vitro

The ability of the LIS-catheters to reduce Fg deposition in vitro was tested for infused medical-grade silicone material and two commercially available urinary catheters, Dover and Bardex with UM versions of each used as controls. The UM- and LIS-catheters were incubated with Fg overnight and assessment of Fg deposition by IF. We found that Fg deposition was reduced in all LIS-catheters, showing ~90% decrease on Dover and ~100% on the Bardex and medical-grade silicone tubing when compared with the corresponding UM controls (*Figure 3A, B*).

Based on previous reports of the biofouling ability of liquid-infused surfaces and our LIS's success in reducing Fg deposition, we tested its ability to prevent microbial surface binding (*Howell et al., 2018*). Our six uropathogens were grown in urine supplemented with BSA at 37°C (*Supplementary file 1*), cultures were normalized in urine, added to UM control and LIS-catheters, incubated under static conditions and quantified via IF. Binding analysis by each of the uropathogens showed that all were able to bind in significantly higher densities to the UM-catheters than to the LIS-catheters (*Figure 3C*). These results further demonstrate the capability of silicone LIS-catheters to reduce not only protein deposition but to also impede microbial colonization.

## Fg deposition and microbial biofilms on catheters was reduced by LIS

Mice were catheterized with either an UM- or LIS-catheter and infected with one of six uropathogens for 24 hr. Bladders and catheters were harvested and assessed for microbial burden by CFU enumeration or fixed for staining. Kidneys, spleens, and hearts were collected to determine microbial burden. We found that mice with LIS-catheters significantly reduced microbial colonization in the bladder and on catheters when compared with UM-catheterized mice regardless of the infecting uropathogen (*Figure 4*). Additionally, colonization was significantly lower in LIS-catheterized mouse kidneys for *P. aeruginosa*, *A. baumannii*, and *E. coli* infections (*Figure 4C, E, G*) and LIS-catheterized mice infected with *E. coli* or *C. albicans* showed significantly less colonization of the spleen (*Figure 4G, K*). *K. pneumoniae* kidney and spleen colonization difference was not statistically significant, however they showed a trend of less colonization and there was significantly less colonization of the heart (*Figure 4I*).

Furthermore, IF imaging and quantification of catheters confirmed decreased Fg deposition and microbial biofilms on LIS-catheters compared to UM (*Figure 4B, D, F, H, J, I* and *Figure 4—figure supplement 1*). These data demonstrate pathogens preferentially bind to Fg, and that the LIS modification successfully reduced Fg deposition (the microbes' binding platform), disrupts uropathogen biofilm formation on catheters, and colonization of the bladder in vivo. Importantly, hematoxylin and eosin (H&E) analysis shows the LI-catheter does not exacerbate bladder inflammation regardless of the presence of infection or not (*Figure 5A–G*), an important factor to account for when developing

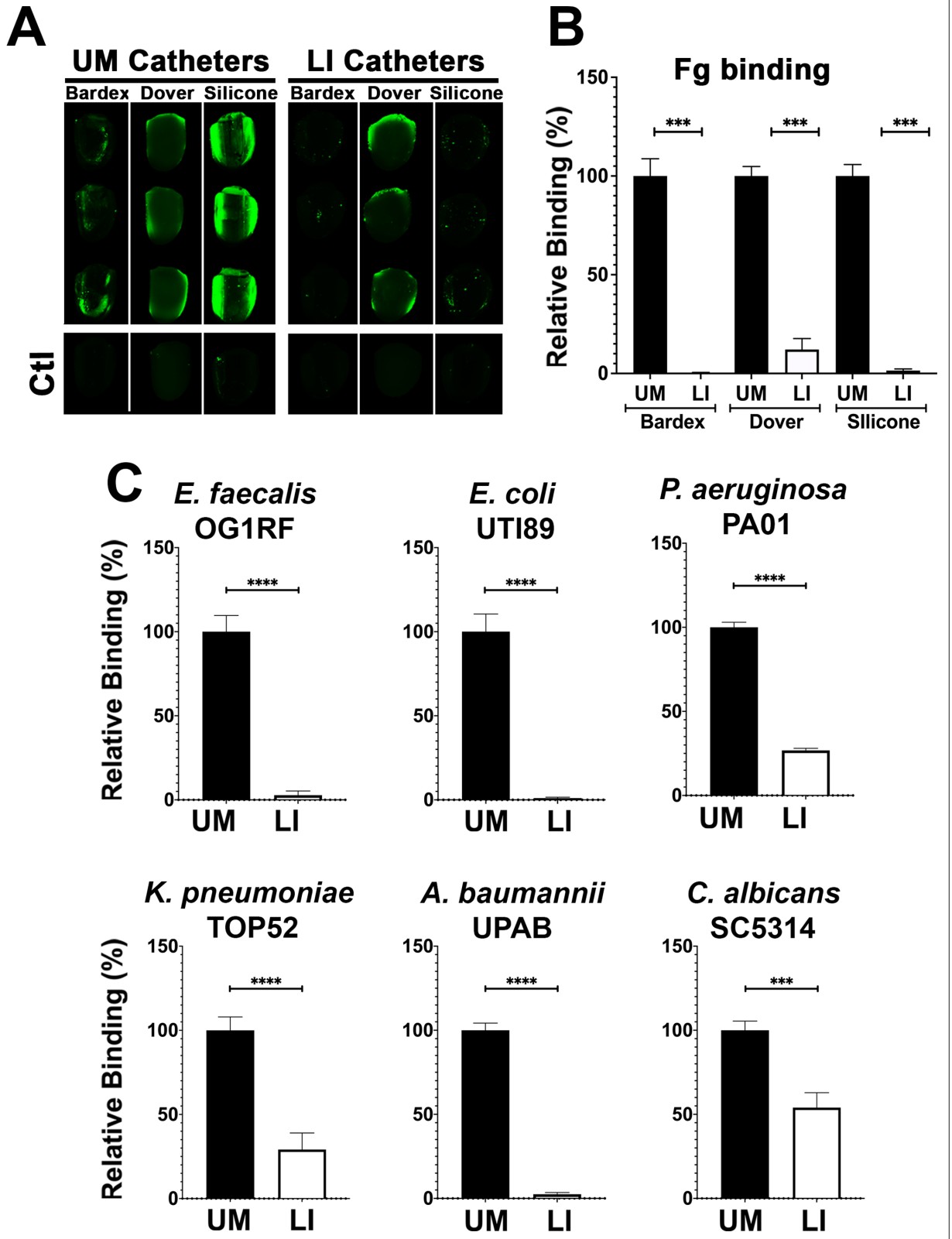

**Figure 3.** Liquid-infused silicone (LIS) modification reduces fibrinogen (Fg) deposition and microbial-binding in vitro. (**A**) Visualization and (**B**) quantification of Fg (green) deposition on unmodified (UM)-catheter material (black bars) and LIS-catheter materials (white bars) by immunofluorescence (IF) staining. Three replicates with $n = 2$–3 each. (**C**) Microbial binding on UM and LIS. For all graphs, error bars show the standard error of the mean (SEM) and each condition had 3 replicates with $n = 3$ each. Differences between groups were tested for significance using the Mann-Whitney U test. ***, $P < 0.0005$; and ****, $P < 0.0001$*.

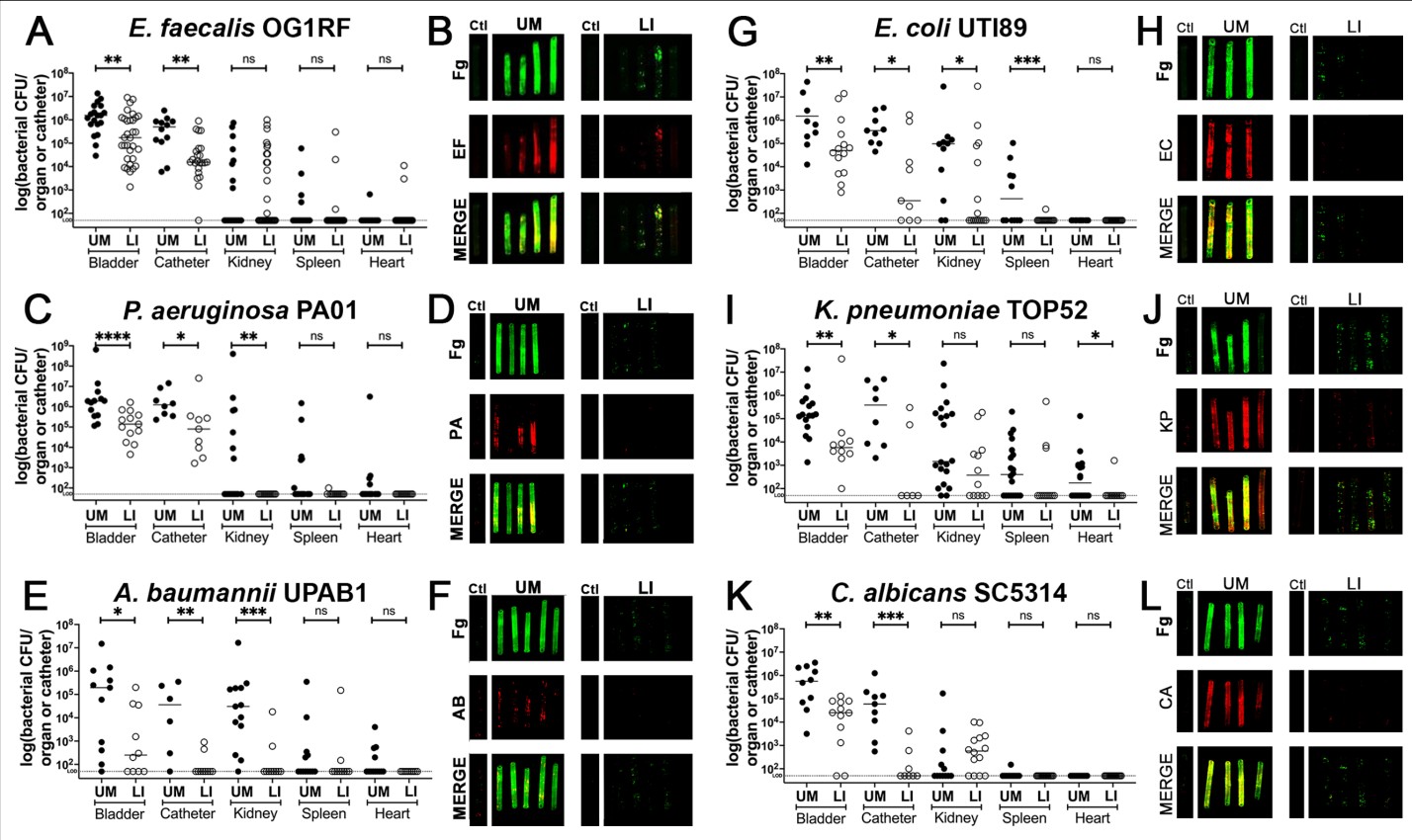

**Figure 4.** In vivo reduction of fibrinogen (Fg) deposition to restrict microbial burden. Mice were catheterized and infected with one of six uropathogens. (**A, C, E, G, I, K**) Organ and catheter CFUs from mice with either an unmodified (UM)-catheter (closed circles) or liquid-infused silicone (LIS)-catheter (open circles) show the dissemination profile of the pathogen. (**B, D, F, H,J,L**) Imaging of catheters for Fg (green), respective uropathogen (red), and a merged image compare deposition on UM-catheters (left) with LIS-catheters (right); nonimplanted catheters as controls. Quantification of microbial colonization and colocalization on the catheters can be found in *Figure 4—figure supplement 1*. All animal studies for CFUs, catheter and bladder imaging had at least 10 animals per strain and catheter type. Differences between groups were tested for significance using the Mann-Whitney U test. *, P < 0.05; **, P < 0.005; ***, P < 0.0005; and ****, P < 0.0001.

The online version of this article includes the following figure supplement(s) for figure 4:

**Figure supplement 1.** Pathogen predilection for fibrinogen (Fg).

a new medical device. In fact, for some pathogens, the LI-catheter results in less inflammation than bladders catheterized with an UM-catheter (*Figure 5B, C, G*). Furthermore, we examine Fg presence, uropathogen colonization, and neutrophil recruitment in UM- and LIS-catheterized and infected bladders by IF microscopy. This analysis revealed a reduction of microbial colonization as well as decreased neutrophil recruitment (*Figure 5H–M*).

## LIS modification reduces protein deposition on catheters in CAUTI mouse model of *E. faecalis*

A quantitative-proteomics comparison was performed to identify proteins deposited on UM- and LIS-catheters retrieved 24 hpi with *E. faecalis*. Harvested catheters were prepared and protease digested with trypsin as in *Zougman et al., 2014*. *n*LC–MS/MS was performed in technical duplicate and label-free-proteomics (LFQ) processed as in Cox and Mann, 865 proteins were identified at a 1% False Discovery Rate (FDR) (*Cox and Mann, 2008*). Total abundance of protein was significantly reduced in LIS- vs UM-catheters (*Figure 6A* and *Supplementary file 2*). Additionally, abundance of Fg and over 130 other proteins significantly decreased while only three proteins showed a significant increase (UDP-glucose 6-dehydrogenase, filamin-B, and proteasome subunit beta type-5) (*Figure 6B* and *Supplementary file 2*). These data further demonstrate that the LIS modification not only reduced Fg

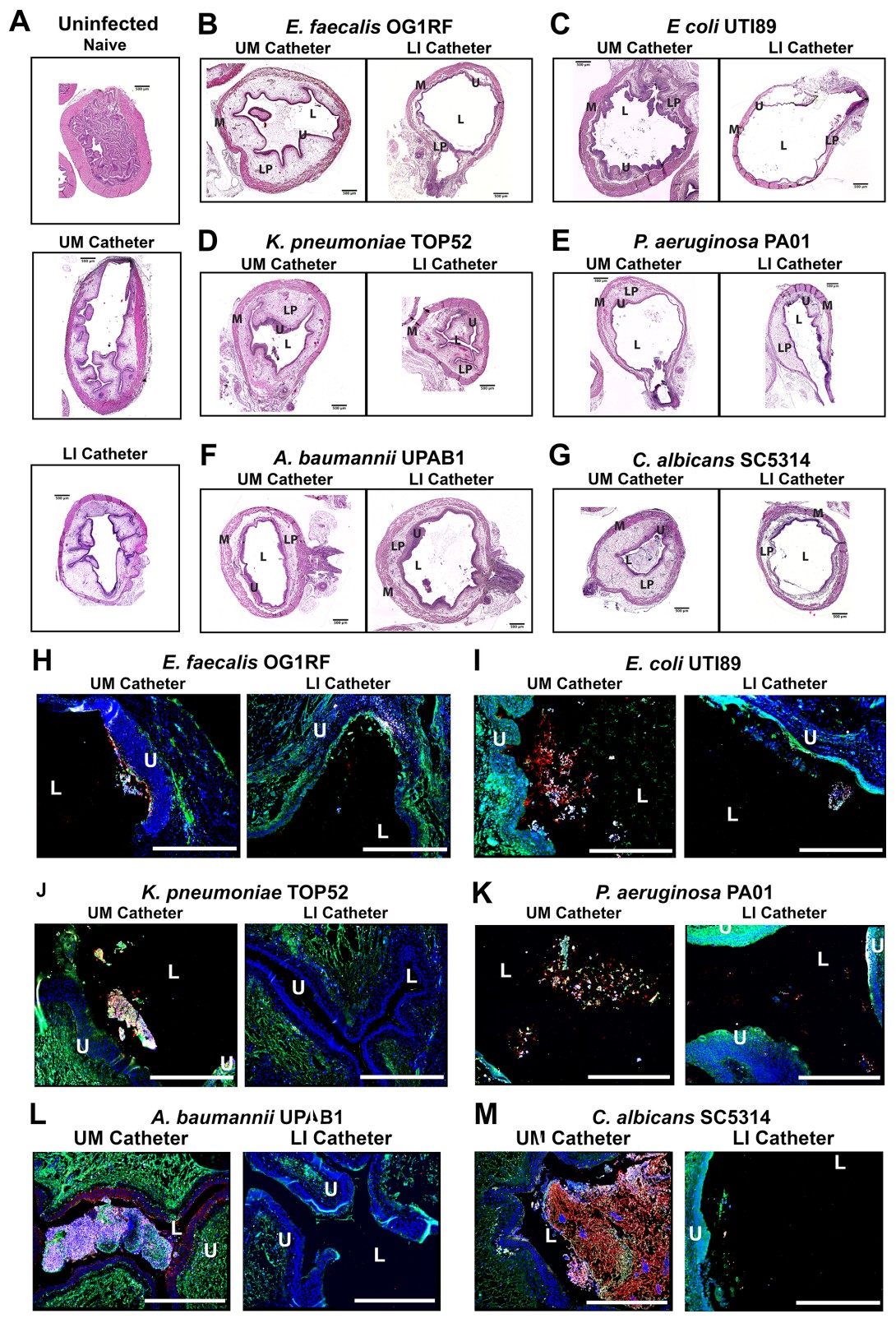

**Figure 5.** Liquid-infused silicone (LIS)-catheters reduce bladder colonization and inflammation. Mice were catheterized and inoculated one of six strains. (**A**) Naive or bladders catheterized with an unmodified (UM)- or LIS-catheter were uninfected controls. (**B–G**) Bladder sections were stained with hematoxylin and eosin (H&E) to compare inflammation from UM-catheters (left) and LIS-catheters (right). (**C–M**) ×20 images of immunofluorescence (IF) stained bladders catheterized with an UM-catheter (left panels) or a LIS-catheter (right panels). Bladders stained for nuclei (blue), fibrinogen (Fg; green),

*Figure 5 continued on next page*

*Figure 5 continued*

respective uropathogens (red), and neutrophils (white). The urothelial/lumen boundaries are outlined in white dotted lines and labeled U (urothelium) and L (lumen) and all scale bars are 500 µm. Montages can be found in *Figure 5—figure supplement 1*.

The online version of this article includes the following figure supplement(s) for figure 5:

**Figure supplement 1.** Montage of *Figure 5* merged images.

deposition but also a wide variety of host proteins, which could play a role in microbial colonization and biofilm formation as demonstrated earlier with BSA (*Figure 2C, E*).

## Discussion

This is the first study to show a diverse set of uropathogens including gram-negative, gram-positive, and fungal species interact with Fg to more effectively bind to silicone urinary catheter surfaces. Furthermore, we found that by disrupting Fg deposition with LIS-catheters we reduced the ability of uropathogens to bind and colonize the catheter surface and bladder in an in vivo CAUTI mouse model. Moreover, LIS also reduced dissemination of *E. coli*, *P. aeruginosa*, and *A. baumannii* into the kidneys and other organs. Furthermore, LIS-catheters did not increase inflammation and for half of the pathogens inflammation was reduced. Finally, the deposition of other host-secreted proteins on LIS-catheters was around 6.5-fold less then UM-catheters. Together, these findings indicate that catheters made using LIS are a promising new antibiotic sparring approach for reducing or even preventing CAUTIs by interfering with protein deposition (*Figure 7*).

Pathogen–Fg interaction has been shown to be important for both *E. faecalis* and *S. aureus* during human and mouse CAUTI (*Flores-Mireles et al., 2016b*; *Walker et al., 2017*). Binding to Fg is critical

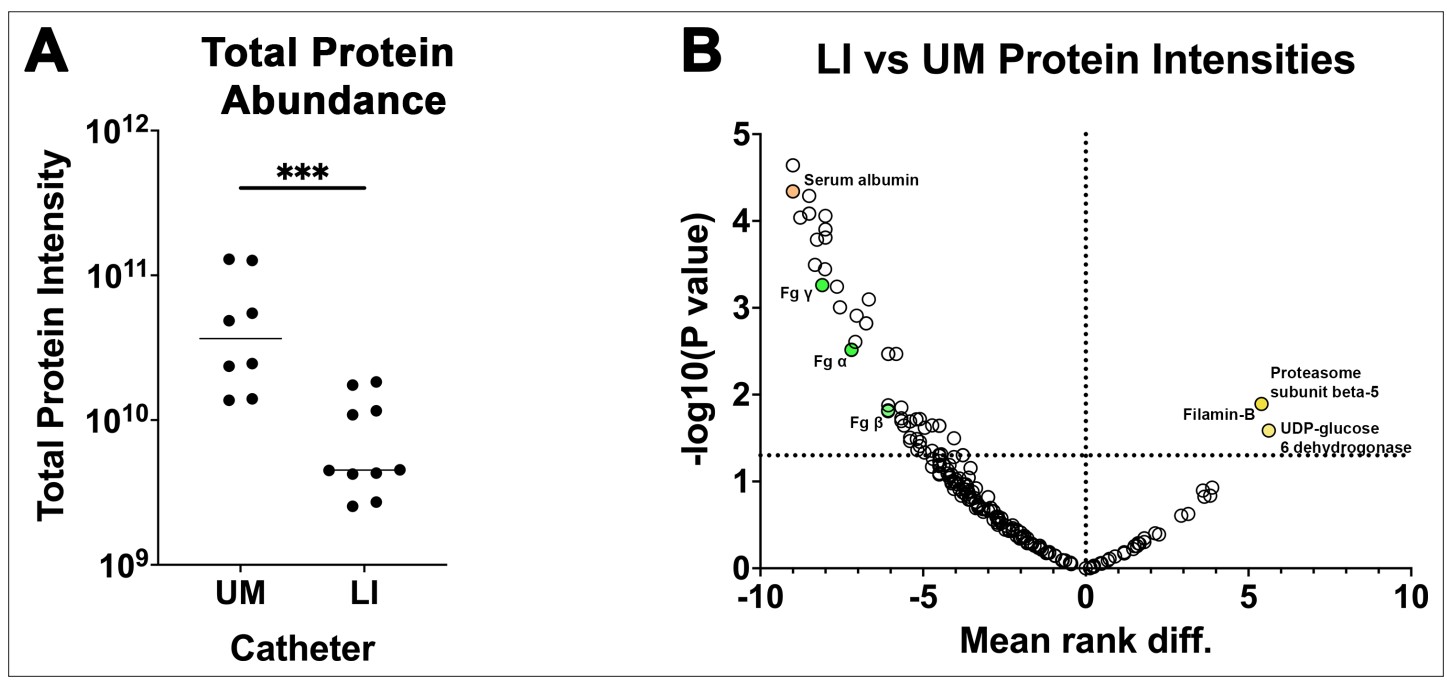

**Figure 6.** Liquid-infused silicone (LIS)-catheter reduces host-protein deposition in vivo. A subset of unmodified (UM)- and LIS-catheters taken from mice 24 hpi with *E. faecalis* were assessed for protein deposition via mass spectrometry four UM-catheters and five LIS-catheters were used. (**A**) Intensities of the 95% most abundant proteins were summed in a total proteome approach and compared between the UM- and LIS-catheter groups. (**B**) A volcano plot for a subset of proteins. Negative mean rank difference indicates less protein on the LIS-catheter then on the UM-catheter and a significant difference is a −log10(p value) over 1.3. The fibrinogen (Fg) chains (α, β, and γ) are highlighted in green, serum albumin in orange, UDP-glucose 6-dehydrogenase, filamin-B, and proteasome subunit beta type-5 in yellow. Differences between groups were tested for significance using the Mann-Whitney U test. ***, P < 0.0005.

The online version of this article includes the following source data for figure 6:

**Source data 1.** Source data for small datasets.

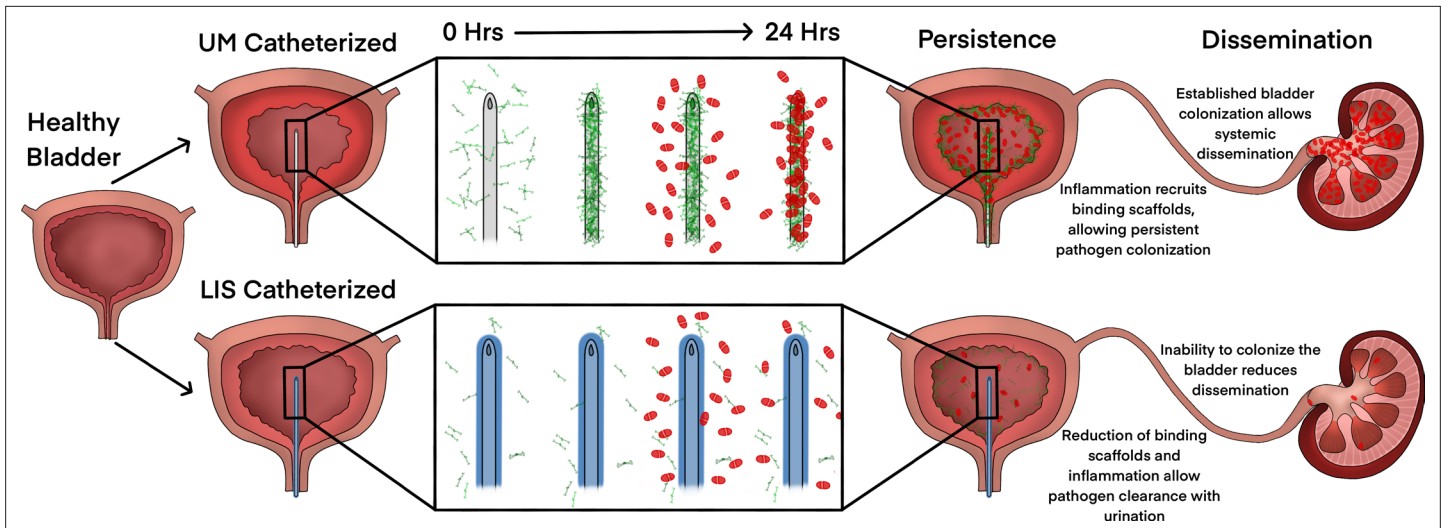

**Figure 7.** Liquid-infused silicone (LIS)-catheter reduces bladder inflammation, incidence of catheter-associated urinary tract infection (CAUTI), and dissemination. Urinary catheter-induced inflammation promotes the release of fibrinogen (Fg) into the bladder to heal physical damage. Consequently, this Fg is deposited onto the catheter creating a scaffold for incoming pathogens to bind, establish infection, and promote systemic dissemination. However, catheterization with a LIS-catheter reduces Fg deposition onto its surface; thus, reducing the availability of a binding scaffolds for incoming pathogens. Consequently, overall bladder colonization and systemic dissemination are reduced making LIS-catheters a strong candidate for CAUTI prevention.

for efficient bladder colonization and biofilm formation on the catheter via protein–protein interaction using EbpA and ClfB adhesins, respectively, and their disruption hinders colonization (*Flores-Mireles et al., 2014*; *Walker et al., 2017*). Gram-negative pathogens, *A. baumannii* and *P. mirabilis*, have shown colocalization with Fg during urinary catheterization (*Gaston et al., 2020*); however, the bacterial factors and any mode of interaction have not been described. While interaction of *E. coli* and *K. pneumoniae* with Fg during CAUTI has not been described previously, pathogenesis during urinary tract infection (UTI) (no catheterization) has been extensively studied. These studies show that type 1 pili, a chaperone–usher pathway (CUP) pili, allows them to colonize the bladder urothelium by binding to mannosylated receptors on the urothelial surface through the tip adhesin FimH (*Klein and Hultgren, 2020*). Furthermore, other CUP pili including the P pili, important for pyelonephritis, and the Fml pilus, important for colonizing inflamed bladder urothelium, bind specifically to sugar residues Galα1–4Gal in glycolipids and Gal(β1–3)GalNAc in glycoproteins, respectively (*Campoccia et al., 2013*). Interestingly, Fg is highly glycosylated, containing a wide variety of sugar residues including mannose, *N*-acetyl glucosamine, fucose, galactose, and *N*-acetylneuraminic acid (*Adamczyk et al., 2013*). Therefore, the glycosylation in Fg may be recognized by CUP pili, allowing pathogens expressing CPU adhesins to become established and colonize the bladder. Furthermore, *A. baumannii* CUP1 and CUP2 pili are essential for CAUTI, this together with *A. baumannii* interactions with Fg in vivo, suggests that these pili may play a role in Fg interaction (*Di Venanzio et al., 2019*). Similarly, *P. aeruginosa* also encodes CUP pili, CupA, CupB, CupC, and CupD, which have shown to be important for biofilm formation (*Mikkelsen et al., 2013*); however, their contribution during CAUTI has yet to be investigated. Furthermore, *C. albicans* has several adhesins, ALS1, ALS3, and ALS9, which have a conserved peptide-binding cavity shown to bind to Fg γ-chain (*Hoyer and Cota, 2016*).

Inhibition of initial uropathogen binding is crucial to reduce colonization and biofilm formation on urinary catheter surfaces and prevent subsequent CAUTI (*Flores-Mireles et al., 2014*; *Flores-Mireles et al., 2016b*). To prevent surface binding, a variety of modified surfaces impregnated with antimicrobial or bacteriostatic compounds have been generated, and have proven to reduce microbial-binding in vitro but not in vivo (*Andersen and Flores-Mireles, 2019*; *Singha et al., 2017*). It is possible that in vitro studies do not efficiently mimic the complexities of the in vivo environment, for example; (1) Growth media: the majority of in vitro studies use laboratory rich or defined culture media, and it has been shown that laboratory media do not recapitulate the catheterized bladder environment that pathogens encounter (*Colomer-Winter et al., 2019*; *Xu et al., 2017*). Specifically, urine culture

conditions have shown to activate different bacterial transcriptional profiles than when cultures are grown in defined media (*Conover et al., 2016*; *Xu et al., 2017*), which may affect microbial persistence and survival. (2) Host factors: host-secreted proteins are released into the bladder due to catheter-induced physical damage and subsequent inflammation. These proteins are deposited on the catheter surface and may hinder the release of antimicrobials or block interaction of antimicrobials with the pathogen if the antimicrobials are tethered to the surface. As has been observed with Fg deposition on human and mouse catheters, that host-protein deposition is not uniform, which may lead to antimicrobial release or pathogen–antimicrobial agent interaction at a subinhibitory concentrations (*Di Venanzio et al., 2019*; *Flores-Mireles et al., 2014*; *Flores-Mireles et al., 2016a*; *Walker et al., 2017*). Consequently, these interactions can contribute to the development of multidrug resistance among uropathogens ('Antibiotic resistance threats in the *Antibiotic resistance threats in the United States, 2019*', 2019).

Based on the role of deposited host proteins in promoting microbial colonization and presence in human CAUTI (*Flores-Mireles et al., 2016a*; *Flores-Mireles et al., 2016b*; *Walker et al., 2017*), antifouling catheter coatings present a better approach to decreasing CAUTI prevalence rather than using biocidal or biostatic compounds, such as antibiotics, that promote resistance (*Campoccia et al., 2013*; *Singha et al., 2017*). Antifouling coatings made from polymers have shown resistance to protein deposition *He et al., 2016*; however, these coatings can become unstable over time and be difficult to produce requiring extensive wet chemistry (*Singha et al., 2017*). Most reports on the use of purely antifouling coatings to combat CAUTI have shown a successful reduction in bacterial colonization in vitro yet have not been successfully tested in vivo (*Andersen and Flores-Mireles, 2019*). This may be partly explained by the fact that many of the antifouling coatings are optimized to target bacterial adhesion, as it is understood that the first stage of biofilm development is bacterial attachment to a surface (*Faustino et al., 2020*). However, in a complex environment such as the in vivo bladder, the first change to the catheter surface is the adhesion of a complex set of host-generated proteins and biological molecules, generally referred to as a conditioning film, which can mask the surface (*Faustino et al., 2020*; *Gaston et al., 2020*; *Scotland et al., 2020*). Yet studies on catheter coatings to-date have rarely focused on the role of the host in infection establishment; namely, the host-secreted proteins. The data we present here suggest that this missing element may at least partly explain the differing results seen for most antifouling catheter treatments in vitro vs in vivo.

This study used clinically relevant silicone oil to create a simple to make LIS that was not only microbial resistant but also protein resistant, filling in the missing link between in vitro and in vivo work, the conditioning film. Interestingly, despite the increased weight, length, inner diameter, and outer diameterof the LIS mouse catheter by full silicone infusion, the LIS-catheters did not exacerbated the inflammation response. This findings are consistent with previous a in vivo work showing reduced fibrous capsule formation in implants coated with a liquid layer, suggesting that the use of a liquid surface may convey additional anti-inflammatory benefits (*Chen et al., 2017*).

Our mouse model of CAUTI has shown to faithfully recapitulate the pathophysiology of human CAUTI, showing urinary catheterization induces inflammation and contribution of Fg to biofilm formation on catheters recovered from humans suffering of CAUTI (*Delnay et al., 1999*; *Flores-Mireles et al., 2014*; *Flores-Mireles et al., 2016a*; *Flores-Mireles et al., 2016b*; *Glahn et al., 1988*; *Peychl and Zalud, 2008*; *Walker et al., 2017*). This suggests that our findings have the potential to be translated for prevention and management of human CAUTI. However, for successful translation of this technology into humans, we need to further understand the effects of silicone oil on the immune response to infection as well as bladder tissue and better understand the impact the swelling of the material will have on manufacturing these catheters.

A deeper understanding of the pathogenesis of CAUTI is critical to moving beyond current developmental roadblocks and create more efficient intervention strategies. Here, we have shown that that infusion of silicone with an immiscible liquid coating significantly decreases Fg deposition and microbial binding by using in vitro conditions that more thoroughly recapitulate the catheterized bladder environment. Importantly, our in vitro results were confirmed in vivo using our established mouse model of CAUTI. Our data showed that LIS-catheters are refractory to bacterial colonization without targeting microbial survival, which often leads to antimicrobial resistance. This study has found that protein deposition on urinary catheters is the Achilles heel of CAUTI pathogens, disrupting pathogen–Fg interaction represent a vulnerability that could be exploited. Thus, LIS-catheters hold tremendous

potential for the development of lasting and effective CAUTI treatments. These types of technologies are desperately needed to achieve better public health by decreasing healthcare-associated infections and promoting long-term wellness.

# Materials and methods

**Key resources table**

| Reagent type (species) or resource | Designation | Source or reference | Identifiers | Additional information |
|---|---|---|---|---|
| Strain, strain background (*Enterococcus faecalis*) | OG1RF | ATCC | 47,077 | |
| Strain, strain background (*Escherichia coli*) | UTI89 | Obtained from Dr, Scott Hultgren lab | | |
| Strain, strain background (*Pseudomonas aeruginosa*) | PA01 | ATCC | BAA-47 | |
| Strain, strain background (*Klebsiella pneumoniae*) | TOP52 1721 | Obtained from Dr. Scott Hultgren lab | | |
| Strain, strain background (*Acetobacter baumannii*) | UPAB1 + CUP1,2 | Obtained from Dr Mario Feldman lab **Di Venanzio et al., 2019** | | |
| Strain, strain background (*Candida albicans*) | SC5314 | ATCC | MYA-2876 | |
| Biological sample (*Homosapian female*) | Urine | This study | | IRB #19-04-5273 |
| Antibody | (Goat polyclonal) antifibrinogen | Sigma-Aldrich | Cat# F8512, RRID:AB_259765 | 1:1000 |
| Antibody | (Rabbit polyclonal) anti-strep group d | From Dr. Scott Hultgren lab **Flores-Mireles et al., 2014** | | 1:1000 (in vitro) 1:500 (IHC) |
| Antibody | (Rabbit polyclonal) anti-*E. coli* serotype O/K | Invitrogen | Cat# PA1-25636 RRID:AB_780488 | 1:1000 (in vitro) 1:500 (IHC) |
| Antibody | (Rabbit polyclonal) anti *P. aeruginosa* | Invitrogen | Cat# PA173117 RRID:AB_1018279 | 1:1000 (in vitro) 1:500 (IHC) |
| Antibody | (Rabbit polyclonal) anti-*K. pneumoniae* polyclonal | Thermo Scientific | Cat# PA17226 RRID:AB_559816 | 1:1000 (in vitro) 1:500 (IHC) |
| Antibody | (Rabbit polyclonal) anti *A. baumannii* | **Di Venanzio et al., 2019** | | 1:1000 (in vitro) 1:500 (IHC) |
| Antibody | (Rabbit polyclonal) anti *C. albicans* | Thermo Fisher Scientific | Cat# PA1-27158 RRID:AB_779500 | 1:1000 (in vitro) 1:500 (IHC) |
| Antibody | (Rat polyclonal) anti-Ly6G | BioLegend | Cat# 127602 RRID:AB_1089180 | 1:500 |
| Antibody | (Donkey polyclonal) anti-goat 800CW | LI-COR Biosciences | Cat# 926-32214, RRID:AB_621846 | 1:5000 |
| Antibody | (Donkey polyclonal) anti-rabbit 680RD | LI-COR Biosciences | Cat# 926-68073, RRID:AB_10954442 | 1:5000 |
| Antibody | (Donkey polyclonal) anti-rat 680 | Thermo Fisher | Cat# A-21472, RRID:AB_2535875 | 1:500 |
| Antibody | (Donkey polyclonal) anti-goat 488 | Thermo Fisher | Cat# A11055 RRID:AB_2534102 | 1:500 |
| Antibody | (Donkey polyclonal) anti-rabbit 550 | Thermo Fisher | Cat# A31572 RRID:AB_162543 | 1:500 |
| Peptide, recombinant protein | Fibrinogen | Enzyme Research Laboratories | Cat# FIB 3 | Adjusted to 150 µg/ml in PBS |
| Software, algorithm | Zeiss pro Software | Carl Zeiss Microscopy | | |

*Continued on next page*

*Continued*

| Reagent type (species) or resource | Designation | Source or reference | Identifiers | Additional information |
|---|---|---|---|---|
| Software, algorithm | Image studio software | Licor Biosciences | | |
| Other | Silicone oil | Gelest | 63148-62-9 | 20 cst |

## Mouse infection models

Mice used in this study were ~6-week-old female wild-type C57BL/6 mice purchased from Jackson Laboratory and The National Institute of Cancer Research. Mice were subjected to transurethral implantation and inoculated as previously described (*Conover et al., 2015*). Briefly, mice were anesthetized by inhalation of isoflurane and implanted with a 6-mm-long UM-silicone or LIS-catheter. Mice were infected immediately following catheter implantation with 50 µl of ~2 × 10$^7$ CFU/ml in phosphate-buffered saline (PBS) introduced into the bladder lumen by transurethral inoculation (unless otherwise noted [*Supplementary file 1*]). For all mouse experiments microbes were grown in their corresponding media (*Supplementary file 1*). To harvest the catheters and organs, mice were sacrificed at 24 hpi by cervical dislocation after anesthesia inhalation; the silicone catheter, bladder, kidneys, heart, and spleen were aseptically harvested. Catheters were either subjected to sonication (Branson, Ultrasonic Bath) for CFU enumeration analysis, fixed for imaging via standard IF described below, or sent for proteomic analysis as described below using nonimplanted catheters as controls for all assays. Bladders for IF and histology were fixed and processed as described below. Kidneys, spleens, and hearts were all used for CFU analysis. The University of Notre Dame Institutional Animal Care and Use Committee approved all mouse infections and procedures as part of protocol number 18-08-4792MD. All animal care was consistent with the Guide for the Care and Use of Laboratory Animals from the National Research Council.

## Bladder Immunohistochemistry (IHC) and H&E staining of mouse bladders

Mouse bladders were fixed in 10% neutralized formalin (Leica) overnight, before being processed and sectioned by ND CORE. Staining was done as previously described (*Walker et al., 2017*). Briefly, bladder sections were deparaffinized, rehydrated, and rinsed with water. Antigen retrieval was accomplished by boiling the samples in Na-citrate, washing in water, and then incubating in PBS three times. Sections were then blocked (1× PBS, 1.5% BSA, 0.1% sodium azide), washed in PBS, and incubated with appropriate primary antibodies overnight at 4°C. Next, sections were washed with PBS, incubated with secondary antibodies for 2 hr at RT, and washed once more in PBS prior to Hoechst dye staining. H&E stain for light microscopy was done by the CORE facilities at the University of Notre Dame (ND CORE). All imaging was done using a Zeiss inverted light microscope (Carl Zeiss, Axio Observer). Zen Pro (Carl Zeiss, Thornwood, NY) and ImageJ software were used to analyze the images.

## Quantifying catheter colonization and Fg deposition

As previously described (*Colomer-Winter et al., 2019*). Briefly, catheters were fixed with 10% neutralized formalin, blocked, and stained using Goat anti-Fg primary antibody (Sigma) (1:1000) and Rabbit anti-pathogen (*Supplementary file 1*) followed by Donkey anti-Goat IRD800 antibody (Invitrogen) (1:5000) and Donkey anti-Rabbit IRD680 antibody (Invitrogen) (1:5000) secondary. Catheters were then dried over night at 4°C and imaged on an Odyssey Imaging System (LI-COR Biosciences) to examine the infrared signal. Images of the signals (Fg in green and pathogens in red) were analyzed in ImageJ using Pixel color counter (*Gaston et al., 2020*).

## Human urine collection

Human urine was collected and pooled from at least two healthy female donors between 20 and 40 years of age. Donors had no history of kidney disease, diabetes, or recent antibiotic treatment. Urine was sterilized using a 0.22 µm filter (Sigma-Aldrich) and pH adjusted to 6.0–6.5. When supplemented with BSA (VWR Lifesciences), urine was filter sterilized again following BSA addition. All participants signed an informed consent form and protocols were approved by the local Internal Review Board at the University of Notre Dame under study #19-04-5273.

## Microbial growth conditions in supplemented urine

*E. faecalis* and *C. albicans* were grown static for ~5 hr in 5 ml of respective media (*Supplementary file 1*) followed by static overnight culture in human urine supplemented with 20 mg/ml BSA (urine BSA20). *E. coli*, *K. pneumoniae*, *P. mirabilis*, *A. baumanii*, and *P. aeruginosa* were grown 5 hr shaking at 37°C in LB then static in fresh urine BSA for 24 hr then, supplemented into fresh urine BSA for an additional 24 hr static (2 × 24 hr) in urine BSA20. All cultures were washed in PBS (Sigma) three times and resuspended in assay appropriate media.

## Silicone disk preparation

Disks of UM-silicone (Nalgene 50 silicone tubing, Brand Products) or LIS were cut using an 8 mm leather hole punch. UM disks were washed three times in PBS and air dried. LIS disks were stored in filter sterilized silicone oil at RT. Disks were skewered onto needles (BD) to hold them in place and put in 5 ml glass tubes (Thermo Scientific) or placed on the bottom of 96-well plate wells (Fisher Scientific) (UM-silicone only). Plates and glass tubes were UV sterilized for >30 min prior to use.

## Protein-binding assays

Human Fg free from plasminogen and von Willebrand factor (Enzyme Research Laboratory #FB3) was diluted to 150 µg/ml in PBS. 500 µL of 150 µg/ml Fg was added to each disk in glass tubes, sealed, and left over night at 4°C. Disks were then processed according to standard IF procedure as described above (*Colomer-Winter et al., 2019*). Briefly, disks were washed three times in PBS, fixed with 10% neutralized formalin (Leica), blocked, and stained using Goat anti-Fg primary antibody (Sigma) (1:1000) and Donkey anti-Goat IRD800 secondary antibody (Invitrogen) (1:5000). Disks were then dried over night at 4°C and imaged on an Odyssey Imaging System (LI-COR Biosciences) to examine the infrared signal. Intensities for each catheter piece were normalized against a negative control and then made relative to the pieces coated with Fg which was assigned to 100%. Images were processed using Image Studio Software (LI-COR, Lincoln, NE) Microsoft Excel and graphed on GraphPad Prism (GraphPad Software, San Diego, CA).

## Microbial-binding assays

For assessing the effect of protein deposition on microbial binding, 100 µl of 150 µg/ml Human Fg, 100 µl of 150 µg/ml BSA, or 100 µl of PBS were incubated on UM-silicone disks in 96-well plates overnight at 4°C. The following day disks were washed three times with PBS followed by a 2-hr RT incubation in 100 µl of urine containing microbes at a concentration of ~$10^8$ CFU/ml.

For assessing microbial binding to UM-silicone vs LIS, 500 µl of microbe containing media was added to prepared disks in glass tubes. Standard IF procedure was then followed as described above using goat anti-Fg and rabbit anti-microbe primary antibodies (1:1000) (see *Supplementary file 1* for details). Secondary antibodies used were Donkey anti-Goat IRD800 and Donkey anti-Rabbit IRD680 (1:5000). Quantification of binding was done using ImageStudio Software (LI-COR). Intensities for each catheter piece were normalized against a negative control and then made relative to the pieces coated with Fg which was assigned to 100%.

## Silicone and Tygon tube infusion

Five samples of 20-cm Tygon tube (14-171-219, Saint-Gobain Tygon S3 TM 3603 Flexible Tubings, Fisher Scientific, USA) or silicone tube (8060-0030, NalgeneTM 50 Platinum-cured Silicone Tubing, Thermo Scientific, USA) was utilized in weight measurement. Weight of the tubes prior to infusion were measured with an analytical balance (AL204, Analytical Balance, Mettler Toledo, Germany). After the measurement of the initial weights, the tubes were submerged in silicone oil (DMS-T15, polydimethylsiloxane, trimethylsiloxy, 50 cSt, GelestSInc, USA) and weighed at designated time points. For each time point, tubes were removed from the oil with forceps and held vertically for 30 s for the excess silicone oil to flow out of the tube. The bottoms of the tubes were then gently dabbed with Kimwipes (Kimwipe, Kimberly-Clark Corp., USA). After measurement, the tubes were again submerged in silicone oil until the next time point. Tubes were measured every 3 hr for the first 2 days; every 6 hr from days 3 to 6; and every 24 hr from day 6 and onwards. Measurements were taken until data showed no significant increase, and that the plateau trendline consist of at least three data points. Based on these

data, for all the protein- and microbial-binding assays, silicone catheters were submerged in silicone oil for 5 days prior to use to ensure full infusion.

## Mouse catheter infusion

Five samples of 20-cm mouse catheter (SIL 025, RenaSil Silicone Rubber Tubing, Braintree Scientific, Inc, USA) were utilized in weight measurement. Weight of the tubes prior to infusion were measured with an analytical balance. After the measurement of the initial weights, the tubes were submerged in silicone oil for different time points. For each time point, catheters were removed from the oil with forceps and a Kimwipes was immediately pressed against the bottom of the catheters to remove the excess silicone oil via capillary action. After the excess oil was drained, catheters were weighed and placed back into silicone oil to continue with the infusion until the next time point. Catheters were measured every 1 min for the first 5 min of the experiment; every 2 min from 5 to 15 min; and every 5 min from 15 min and onwards. Measurements were taken until weight showed no significant increase, and that the plateau trendline consist of at least three data points. Silicone catheters modified and used in mouse infections were submerged in silicone oil for at least 30 min prior to use.

## Parameter measurement of silicone tube before and after infusion

The length, inner diameter, and outer diameter of silicone tubes were measured before silicone oil infusion and following complete infusion (after incubating with silicone oil for >5 days). All parameters were measured using a digital caliper (06-664-16, Fisherbrand Traceable Digital Calipers, Fisher Scientific, USA). For mouse catheters, length was measured using a digital caliper. For inner and outer diameter measurement, photos of the tube openings of the catheters and a scale of known length were taken and estimated using ImageJ. Percentage weight change of Tygon tube, silicone tube, and mouse catheters were calculated based on the formula below:

$$\frac{\text{Weight of the tube} - \text{Initial weight of the tube}}{\text{Initial weight of the tube}} * 100\%$$

## Proteomic analysis of mouse catheters

Five mice catheterized with a LIS-catheter and four mice were catheterized with an UM-catheter were sacrificed after 24 hr of infection with *E. faecalis* OG1RF. Catheters were harvested and put into 100 µl of sodium dodecyl sulfate (SDS buffer (100 mM Tris–HCl, pH 8.8, 10 mM Dithiothreitol (DTT), and 2% SDS)), then vortexed for 30 s, heated for 5 min at 90°C, sonicated for 30 min and the process repeated once more. Samples were sent to the Mass Spectrometry and Proteomics Facility at Notre Dame (MSPF) for proteomic analysis. Proteins were further reduced in DTT, alkylated and digested with trypsin using Suspension Trap and protocols (*Zougman et al., 2014*). *n*LC–MS–MS/MS was performed essentially as described in *Sanchez et al., 2020* on a Q-Exactive instrument (Thermo). Proteins were identified and quantified using MaxLFQ (Label Free Quantification) within MaxQuant and cutoff at a 1% FDR (*Cox and Mann, 2008*). This generated a total of eight data records from UM-catheters and 10 from LIS-catheters. Data reduction was performed by removing contaminants proteins. Protein abundance for each catheter type was then calculated by summing the LFQ intensity of proteins which comprised 95% of the total abundance on the catheters. Strict filtering criteria of at least two replicates with technical duplication from the UM-catheters and three replicates with technical duplication from the LIS-catheters were required to keep an identification. Abundance of the reduced proteins was plotted using GraphPad Prism. Statistical significance was tested using Mann–Whitney *U*. A volcano plot was created using the ranked mean difference for each protein and −log of calculated p values with alpha = 0.05.

## Statistical analysis

Unless otherwise stated, data from at least three experiments were pooled for each assay. Significance of experimental results was assessed by Mann–Whitney *U* test using GraphPad Prism, version 7.03 (GraphPad Software, San Diego, CA). Significance values on graphs are *p ≤ 0.05, **p ≤ 0.01, ***p ≤ 0.0001, and ****p ≤ .00001.

## Acknowledgements

We thank members of the Flores-Mireles and Howell laboratories for their helpful suggestions and for making this project possible. Special thank you to Dr. Matthew Champion and the mass spectroscopy and proteomics facility at the University of Notre Dame for their insight into proteomic analysis and the ND CORE Facilities for their work on tissue processing. Thank you to Mario Feldman and Gisela Di Venanzio for providing the *A. baumannii* strain and antibodies. This work was funded by the University of Notre Dame Institutional Funds and seed grant FY19SEED6 (to ALFM, AALB, and AM), National Institute of Health grant R01-DK128805 (to ALFM, MJA, CH, and CKF), and National Science Foundation grant no. CBET-2029378 (to CH and CKF).

## Additional information

### Funding

| Funder | Grant reference number | Author |
| --- | --- | --- |
| National Institute of Diabetes and Digestive and Kidney Diseases | R01-DK128805 | Ana L Flores-Mireles<br>Alex Molesan<br>Marissa Jeme Andersen<br>ChunKi Fong<br>Caitlin Howell |
| Division of Chemical, Bioengineering, Environmental, and Transport Systems | CBET-2029378 | ChunKi Fong<br>Caitlin Howell |
| University of Notre Dame | FY19SEED6 | Marissa Jeme Andersen<br>Ana L Flores-Mireles |
| National Science Foundation | CBET-2029378 | Caitlin Howell<br>ChunKi Fong |

The funders had no role in study design, data collection, and interpretation, or the decision to submit the work for publication.

### Author contributions

Marissa Jeme Andersen, Conceptualization, Data curation, Formal analysis, Investigation, Methodology, Validation, Visualization, Writing – original draft, Writing – review and editing; ChunKi Fong, Data curation, Investigation, Writing – original draft; Alyssa Ann La Bella, Alex Molesan, Data curation; Jonathan Jesus Molina, Formal analysis; Matthew M Champion, Formal analysis, Resources; Caitlin Howell, Conceptualization, Funding acquisition, Project administration, Supervision; Ana L Flores-Mireles, Conceptualization, Funding acquisition, Methodology, Project administration, Resources, Supervision, Writing – original draft, Writing – review and editing

### Author ORCIDs

Caitlin Howell http://orcid.org/0000-0002-9345-6642
Ana L Flores-Mireles http://orcid.org/0000-0003-4610-4246

### Ethics

All participants signed an informed consent form and protocols were approved by the local Internal Review Board at the University of Notre Dame under study #19-04-5273.

The University of Notre Dame Institutional Animal Care and Use Committee approved all mouse infections and procedures as part of protocol number 18-08-4792MD. All animal care was consistent with the Guide for the Care and Use of Laboratory Animals from the National Research Council. All catheterization and infection were done under isoflurane anesthesia, sacrifices were done by isoflurane overdose followed by cervical dislocation and every effort was made to minimize suffering.

### Decision letter and Author response

Decision letter https://doi.org/10.7554/eLife.75798.sa1
Author response https://doi.org/10.7554/eLife.75798.sa2

## Additional files

### Supplementary files

• Supplementary file 1. List of proteins found on LI and UM mouse catheters infected with *E. faecalis* OG1RF. The average number of peptides for each protein found on 10 mouse catheters sorted by greatest abundance on the UM-catheter.

• Supplementary file 2. Microbe details. List of microbial strains and their corresponding growth conditions, inoculum concentrations, and antibodies used in this study

• Transparent reporting form

• Source data 1. Proteomic data from UM- and LIS-catheters.

### Data availability

The data that support the findings of this study are available in the source data. RAW and processed MS–MS/MS data are available in the MassIVE public repository, accession MSV000088527. This study did not generate new unique reagents.

The following dataset was generated:

| Author(s) | Year | Dataset title | Dataset URL | Database and Identifier |
|---|---|---|---|---|
| Champion MM, Andersen MJ, Flores-Mireles AL | 2020 | Inhibiting Host Protein Deposition on Urinary catheters Reduces Urinary Tract Infections | https://massive.ucsd.edu/ProteoSAFe/dataset.jsp?accession=MSV000088527 | MassIVE, MSV000088527 |

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
