## [Editor Report]

We assessed your work and were impressed by the number of different experiments and the coherence between them, clearly demonstrating the potential of your work for future prevention of catheter-associated urinary tract infections.

---

## [Decision Letter]

**Decision letter after peer review:**

Thank you for submitting your article "Inhibiting Host Protein Deposition on Urinary Catheters Reduces Urinary Tract Infections" for consideration by *eLife*. Your article has been reviewed by 2 peer reviewers, including Marc J Bonten as the Reviewing Editor and Reviewer #1, and the evaluation has been overseen by a Reviewing Editor and Wendy Garrett as the Senior Editor.

Essential Revisions (for the authors):

Please address the comments made by both reviewers.

*Reviewer #1 (Recommendations for the authors):*

Further fine tuning of the discussion would improve the manuscript.

*Reviewer #2 (Recommendations for the authors):*

First, may I congratulate you on this excellent article. I learned a lot from it and I am very curious to where this will lead. I have some general issues and some textual issues.

General issues:

I think the readability and comprehension can be improved by adding something of a graphical abstract illustrating the steps of your research. Especially for the fast reader, this offers the possibility to understand the study results and implications in a short time.

Regarding the third step: it is insufficiently clear how the LIS catheter that was used for the mice experiments is manufactured exactly. Is this catheter drained in silicone oil before insertion and for how long? And why is the gain in weight over time so important to register? What can you deduce from this? And speaking of the change of dimensions; what could be the impact of this on the development of CA-UTIs? I think it is important to discuss the implications of these findings.

I think the discussion should focus on what the next steps are in research/development of the LIS catheter, towards its use in clinical practice

Textual issues:

Results:

Do you mean 24 hours?

" Mice catheterized and infected with the respective uropathogen were sacrificed at 24 post infection (hpi)."

Maybe rephrase?

"Analysis found the LIS-catheters showed significantly reduced binding of all uropathogens when compared to μm controls"

Discussion:

You indicate that these pili in vivo interact with Fg and therefore they play a role with Fg interaction? "Furthermore, A. baumannii CUP1 and CUP2 pili are essential for CAUTI, this together with its interactions with Fg in vivo, suggests that these pili may play a role in Fg interaction (Di Venanzio et al., 2019)."

---

## [Author Response]

Reviewer #1 (Recommendations for the authors):Further fine tuning of the discussion would improve the manuscript.

We thank the reviewer for their insightful suggestions to help us to improve the manuscript. We have reviewed and edited the Discussion section.

Reviewer #2 (Recommendations for the authors):First, may I congratulate you on this excellent article. I learned a lot from it and I am very curious to where this will lead. I have some general issues and some textual issues.General issues:I think the readability and comprehension can be improved by adding something of a graphical abstract illustrating the steps of your research. Especially for the fast reader, this offers the possibility to understand the study results and implications in a short time.

We thank the Reviewer for this suggestion. We agree with the Reviewer that a graphical abstract will help to convey our findings. We have added it as Figure 7 (lines 539-546), we will leave to the discretion of the editor to keep it in the manuscript.

Regarding the third step: it is insufficiently clear how the LIS catheter that was used for the mice experiments is manufactured exactly. Is this catheter drained in silicone oil before insertion and for how long? And why is the gain in weight over time so important to register? What can you deduce from this?

We thank the Reviewer for pointing out the lack of details on our part. We have added this information in the results (Lines 134-145) and material and methods (Lines 437-449) sections to explain our use of this data further on in later assays. Silicone infusion was done by submerging the catheter in the silicone oil to achieve full saturation as seen by a plateau in the weight of the tube over time (Figure S2 and Figure S2).

And speaking of the change of dimensions; what could be the impact of this on the development of CA-UTIs? I think it is important to discuss the implications of these findings.

The impact of the swelling of the silicone material during infusion should be taken into account during the manufacturing process. Understanding how much swelling is happening will allow us to predict what dimensions we would need from the beginning of the process to understand the dimensions desired after infusion. The data we have provided in this paper is the beginning of understanding the dimension changes. Furthermore, material flexibility and rigidity compared to current urinary catheter requirements should be assessed. A more in-depth analysis will be done in later publications (Lines 292-293). Importantly, full infusion and thus increased size of the mouse catheters did not exacerbated inflammation and did not promote microbial colonization (Lines 278-285).

I think the discussion should focus on what the next steps are in research/development of the LIS catheter, towards its use in clinical practice

We truly appreciate the reviewer’s feedback to improve our manuscript. We have edited the Discussion section (changes marked in red) and we have now emphasized this point in the Discussion section (Lines 286-295).

Textual issues:Results:Do you mean 24 hours?" Mice catheterized and infected with the respective uropathogen were sacrificed at 24 post infection (hpi)."

Yes, thank you for catching this, it has been changed (Line 101).

Maybe rephrase?"Analysis found the LIS-catheters showed significantly reduced binding of all uropathogens when compared to μm controls".

We have rephrased this to “Binding analysis by each of the uropathogens showed that all were able to bind in significantly higher densities to the UM-catheters than to the LIS-catheters “ (Lines 157-158).

Discussion:You indicate that these pili in vivo interact with Fg and therefore they play a role with Fg interaction? "Furthermore, A. baumannii CUP1 and CUP2 pili are essential for CAUTI, this together with its interactions with Fg in vivo, suggests that these pili may play a role in Fg interaction (Di Venanzio et al., 2019)."

We thank the Reviewer to bring this to our attention. Yes, rereading this we understand the confusion “it” refers to *A. baumannii* not the pili, this has been updated in the manuscript to “…this together with *A. baumannii’s* interaction with Fg in vivo…” (Lines 228-230)n and microbial biofilms on LIS-catheters